# The Prevalence and Risk Factors of Chronic Heart Failure in the Mongolian Population

**DOI:** 10.3390/diagnostics13050999

**Published:** 2023-03-06

**Authors:** Pagmadulam Sukhbaatar, Batzorig Bayartsogt, Ganchimeg Ulziisaikhan, Bolortuul Byambatsogt, Chingerel Khorloo, Burmaa Badrakh, Sumiya Tserendavaa, Naranchimeg Sodovsuren, Mungunchimeg Dagva, Mungun-Ulzii Khurelbaatar, Sodchimeg Tsedensodnom, Bat-Erdene Nyamsuren, Rinchyenkhand Myagmardorj, Tsolmon Unurjargal

**Affiliations:** 1Department of Cardiology, School of Medicine, Mongolian National University of Medical Sciences, Ulaanbaatar 14210, Mongolia; 2Department of Epidemiology and Biostatistics, School of Public Health, Mongolian National University of Medical Sciences, Ulaanbaatar 14210, Mongolia; 3National Cardiovascular Center of Mongolia, The Third State Central Hospital, Ulaanbaatar 16081, Mongolia; 4Department of Communication Skill, Bio-Medical School, Mongolian National University of Medical Sciences, Ulaanbaatar 14210, Mongolia; 5Cardiac Rhythmology Center of the Third State Central Hospital Mongolia, Ulaanbaatar 16081, Mongolia; 6Nobel Pharmaceutical Company, Ulaanbaatar 16100, Mongolia; 7Cardiovascular Department, University Hospital of Mongolian National University of Medical Sciences, Ulaanbaatar 13270, Mongolia

**Keywords:** heart failure, prevalence, risk factor, diagnostic criteria

## Abstract

Background: The prevalence of heart failure in the Mongolian population is unknown. Thus, in this study, we aimed to define the prevalence of heart failure in the Mongolian population and to identify significant risk factors for heart failure among Mongolian adults. Methods: This population-based study included individuals 20 years and older from seven provinces as well as six districts of the capital city of Mongolia. The prevalence of heart failure was based on the European Society of Cardiology diagnostic criteria. Results: In total, 3480 participants were enrolled, of which 1345 (38.6%) participants were males, and the median age was 41.0 years (IQR 30–54 years). The overall prevalence of heart failure was 4.94%. Patients with heart failure had significantly higher body mass index, heart rate, oxygen saturation, respiratory rate, and systolic/diastolic blood pressure than patients without heart failure. In the logistic regression analysis, hypertension (OR 4.855, 95% CI 3.127–7.538), previous myocardial infarction (OR 5.117, 95% CI 3.040–9.350), and valvular heart disease (OR 3.872, 95% CI 2.112–7.099) were significantly correlated with heart failure. Conclusions: This is the first report on the prevalence of heart failure in the Mongolian population. Among the cardiovascular diseases, hypertension, old myocardial infarction, and valvular heart disease were identified as the three foremost risk factors in the development of heart failure.

## 1. Introduction

Heart failure (HF) is one of the major public health concerns with an increasing incidence over the years and it remains to be one of the leading causes of mortality among cardiovascular (CV) diseases [1]. As of 2020, 64.3 million people were suffering from chronic HF worldwide [2]. In recent years, chronic HF has become more prevalent because of an aging population, increased cardiovascular risk factors caused by economic trends promoting unhealthy lifestyle behaviors, and modern therapeutic advances that have been extending the lifespan of patients with CV diseases. In the future, the number of patients with HF will continue to rise globally due to the above-mentioned reasons as well as the rise in related comorbidities [3].

HF is becoming an epidemic which has significant epidemiological variations among different geographic regions and countries [4]. Geographically, the prevalence of HF varies; the Middle East, North Africa, and Central Europe have the highest prevalence rates, whereas Southeast Asia and Eastern Europe have lower rates. In the United States, the prevalence of HF has been reported to range from 2.4 to 3.0%; in European countries such as Italy, England, France, and Germany it is between 1.2 and 3.9% of the total population; in some Asian countries (Indonesia and Taiwan) the prevalence of chronic HF is 4–6% [5].

According to the EPICA (Epidemiology of Heart Failure and Learning) research conducted in late 1990 in Portugal, the prevalence of chronic HF was 1.36% in the age group 25–49 years, 2.93% in the age group 50–59 years, 7.63% in the age group 60–69 years, 12.67% in the age group 70–79 years, and 16.14% for those aged over 80 years [6]. The Rotterdam Study revealed that the prevalence of symptomatic HF increased from 0.5% in participants aged 55–64 years to over 14% in participants aged between 85 and 94 years [7]. Chronic HF prevalence increased from 0.7% in the relatively younger age group (45–54 years) to more than 8% in participants over the age of 74 years, according to a population-based study from the USA [8]. Moreover, HF is no longer a disease only for elders, since the prevalence of HF is also rising in younger individuals [9,10].

The prevalence of chronic HF has been reported to be similar between the genders, although there are differences in the characteristics between women and men with HF. Among people aged 65–85 years, the incidence rate of HF in men roughly doubles, whereas in women the HF incidence rate triples. In female patients, HF tends to develop later in life and they have a longer life expectancy compared to men [11].

According to studies from the USA, the main risk factors for HF development include ischemic heart disease, hypertension, diabetes mellitus, older age (>65 years), and obesity [12]. Study results from European countries have also denoted similar risk factors but added smoking as a main risk factor for HF [13]. Interestingly, rheumatic heart disease along with coronary heart disease and hypertension are the most common causes of HF in South Asian developing countries [14]. Additionally, poorly controlled diabetes (HbA1c ≥ 8%), uncontrolled hypertension (SBP ≥ 160), and severe obesity (BMI ≥ 35 kg/m^2^) are the main risk factors contributing to HF incidence [15]. 

In Mongolia, the prevalence of CV diseases has increased 2.5-fold during the past decade and has become a significant concern of the health sector [16]. Identification of prevalence and risk factors for HF will support development of strategies for early detection and prevention [17]; however, there is no comprehensive study on the prevalence and risk factors of HF in Mongolia. The aim of this study is to determine the prevalence and risk factors of HF among the adult population of Mongolia.

## 2. Materials and Methods

### 2.1. Data Source and Sampling

This population-based cross-sectional study was based on subjects attending primary health care centers and was conducted between January and May 2022. Since Mongolia is one of the least densely populated countries in the world with a certain diversity among its population, we used cluster combined with stratified and three-stage random sampling. Based on their geographical region, the country was divided into four distinct sampling zones. The sampling zones included selected districts from the capital city Ulaanbaatar and 7 provinces from the Western, Eastern, Central, and Mountainous regions of the country (Figure 1).

#### 2.1.1. Study Sample Size

The sample size was calculated based on the total population aged 20 years and older (*n* = 2,157,011) and an average prevalence from previous international studies (1.5%) assuming a 95% confidence interval (Z = 1.96) with a 2% acceptable margin of error (e = 0.02), which gave a sample size of 3600 subjects. 

#### 2.1.2. Sample Selection

Study clusters (*n* = 75) and subjects were randomly selected from 7 provinces in the Western, Mountain, Eastern, and Central regions according to geographical zoning and 6 districts of the Ulaanbaatar city. At each primary health care center, the subjects were enrolled in this study by using systematic sampling and were stratified into 10-year-interval age groups. The target sample size including 3600 subjects composed of 900 subjects in the 20–29 age group, 900 subjects in the 30–39 age group, 675 subjects in the 40–49 age group, 600 subjects in the 50–59 age group, 375 subjects in the 60–69 age group, and 150 subjects in the 70 years and over age group. Because the exclusion criterion was subjects with incomplete data, 120 subjects (3.3%) were excluded from the final analysis.

### 2.2. Data Collection

We performed quantitative survey methodology using standard questionnaires. Prior to the data collection, all research staff were provided with detailed instructions and trained for conducting interviews using study questionnaires. The questionnaire included subject’s demographics, social characteristics, presence of CV risk factors, comorbidities, and HF-related symptoms. Educational level was divided into 3 groups, low, medium, and high. Marital status was categorized into 2 groups including married or cohabiting and divorced or single. Lifestyle characteristics such as smoking and alcohol consumption were classified as dichotomous variables: smoker or non-smokers, never or normal/abnormal use of alcohol. 

Diabetes mellitus was defined as self-reported physician-diagnosed diabetes and/or use of insulin and/or oral hypoglycemic medications. Coronary disease was defined as a prior myocardial infarction or revascularization (coronary bypass surgery or angioplasty).

Physical examinations were performed by well-trained physicians in order to identify HF-related signs. Blood pressure, heart rate, oxygen saturation, respiratory rate, and weight were measured by physicians. Body mass index (BMI, kg/m^2^) was calculated by dividing the weight (kg) and height (m^2^). Obesity was defined as a BMI of 30.0 kg/m^2^ or greater. Hypertension was defined by a physician’s diagnosis, systolic blood pressure ≥130 mmHg or diastolic blood pressure ≥80 mmHg, or use of antihypertensive medication. 

We defined atrial fibrillation as a history or the presence of atrial fibrillation on an electrocardiography. CKD was defined as kidney damage or glomerular filtration rate (GFR) <60 mL/min/1.73 m^2^. Anemia was defined as serum hemoglobin levels <13.0 g/dL (<130 g/L) for men and <12.0 g/dL (<120 g/L) for women. The International Classification of Disease (ICD) 10 codes was used for the following comorbidities: COPD(J44), sleep apnea (G47.3), and thyroid disorders (E03 and E05). 

Heart failure was defined as a syndrome recognized by the physician based on symptoms of exercise intolerance, signs of fluid retention, and response to therapy, according to the Guidelines of the ESC Working Group on Heart Failure. In our study, a diagnosis of chronic HF was based on the ESC clinical diagnostic criteria, including if the participant had chronic HF-related symptoms both at rest and during exercise (breathlessness, ankle swelling, and fatigue) or chronic HF-related signs (peripheral oedema, hepatomegaly, neck vein distention, third heart sound (S3) gallop rhythm, and pulmonary crepitations) and, in cases where there was doubt, the patient’s response to diuretic treatment. An HF diagnosis was considered when the first and second criteria were both met [18].

### 2.3. Statistical Analysis

Patients’ demographic characteristics and clinical variables were analyzed in the whole sample using descriptive statistics. Continuous variables were expressed as means ± standard deviations (for normal distribution) or medians with interquartile range (non-normal distribution). Categorical variables were shown as absolute numbers and percentages. The distribution of normality was based on visual assessment of a histogram and the Kolmogorov–Smirnov test. Categorical data were compared using a chi-square test. while continuous variables were compared using an independent sample *t*-test. Correlations between cardiovascular risk factors and both HF and non-HF groups were assessed using the Pearson’s correlation coefficient. In addition, a logistic regression analysis was performed to calculate the odds ratio to assess associations between risk factors and covariates. Statistical significance was considered for two-sided *p*-values less than <0.05. The Statistical Package for the Social Sciences (SPSS 24.0) was used for data analysis.

## 3. Results

In total, 3480 patients aged 18–87 years were enrolled in this study, of which 1345 (39%) were men, and the median age was 41.0 years (IQR 30–54 years). Demographic characteristics of the study population are summarized according to gender in Table 1. Compared to men, women had significantly higher education and intellectual labor than men. Age groups, marital status, and proportion of population in the main administrative groups were comparable in terms of gender.

The prevalence of CV risk factors according to age groups are included in Table 2. The frequencies of some risk factors increased along with an increase in age group; hypertension, diabetes mellitus, and CAD were more prevalent in patients aged 70 years and over. Whereas middle-aged patients (40–49 and 50–59 years, some included 60–69 years) had significantly higher prevalences of the remaining risk factors including valvular heart disease (3%, 5%, and 4%), abnormal alcohol consumption (11% and 13%), smoking (24% and 23%), and obesity (30%, 29%, and 30%).

Comparing the above-mentioned risk factors and comorbidities according to the main administrative groups, in the urban population there was significantly more smoking (22% vs. 19%) and diabetes mellitus (7% vs. 5%) than in the rural population (Table 3), while in the rural population there was significantly more abnormal alcohol consumption (10% vs. 7%) and obesity (25% vs. 21%). 

The overall prevalence of chronic HF was 4.94% for the total study population based on the ESC diagnostic criteria for HF. The prevalence of chronic HF strongly increased with age; it was 0.7% for the 20–29 year age group, while its frequency was 21.0% for the 70–87-year age group (Figure 2).

Figure 3 shows the age-specific prevalence of overall HF for the 10-year-interval age groups in men and women. The prevalence of heart failure increased from 1.2% for men aged 20–29 years to 17.7% for men aged ≥70 years. For women, the prevalence increased from 0.5% in the lowest age group to 23.3% in the highest age group.

In the 40–49 and 50–59 year age groups, men and women showed comparable point prevalences. 

The prevalences of HF in the urban and rural populations are shown in Figure 4. The prevalences of HF for males and females in the rural population were higher than those in the urban population.

The demographic and social characteristics of the study participants are shown in Table 4. The patients with HF were significantly older (median age 57 years), fewer had higher level education (18% vs. 36%), and more had low level education (30% vs. 13%) and were unemployed (58% vs. 40%) compared to subjects without HF. The remaining variables including sex and marital status were comparable. Demographic and social characteristics of the study participants are shown in Table 4. 

For the logistics regression analysis, cardiovascular risk factors were included for analyzing the correlations between the variables and HF (Table 5). Among the cardiovascular risk factors, CAD, hypertension, valvular heart disease, abnormal alcohol consumption, and obesity significantly increased the risk of HF. 

There were significant differences between the non-HF and HF groups regarding cardiovascular risk factors and clinical characteristics based on a physical examination (Table 6). Patients with HF experienced significantly more cardiovascular risk factors. The physical examination for HF showed that patients with HF, as compared with patients without HF, had significantly higher body mass index (28.8 kg/m^2^ vs. 26.2 kg/m^2^), heart rate (84.3 bpm vs. 79.6 bpm), respiratory rate (19.2 vs. 18.2), and systolic (137.7 mmHg vs. 121.2 mmHg) and diastolic blood pressure (86.2 mmHg vs. 78.3 mmHg), and lower oxygen saturation (95.0% vs. 96.2%).

There were significant differences between the non-HF and HF groups regarding use of medications (Table 7). Diuretics were the most used medication, followed by renin-angiotensin system (RAAS) inhibitors and β-blockers.

Atrial fibrillation (24%), sleep apnea (20%), and COPD (16%) were common comorbidities in patients with HF (Figure 5).

## 4. Discussion

First, this is the first study that revealed the prevalence of HF including both urban and rural populations using different clusters. Secondly, we found that Mongolian patients with HF had significantly higher frequencies of comorbidities and risk factors and poorer physical characteristics. Thirdly, hypertension, coronary heart disease, and valvular heart disease were leading CV causes of HF in Mongolian patients.

Based on our study results, the prevalence of HF (4.94%) in Mongolian adults is higher than that reported by studies from USA, some European countries (such as Italy, England, France, and Germany), some Asian countries (such as China and Japan), relatively comparable to Singapore, and lower than the prevalence in Malaysia [6,19,20,21,22]. The present study findings suggest that the reason for the prevalence of HF was higher in the rural population than in the urban population, could be explained by disparities in economic levels (types of occupation), lifestyles (abnormal alcohol usage and obesity), education levels (*p* < 0.0001), and clinical conditions between urban and rural areas. These findings are consistent with those of a study in India [23]. 

Moreover, our findings show that unemployed and low education increase the risk of HF compared to participants without HF. A recent meta-analysis of 11 studies found that low socioeconomic status assessed by all common measures (education, income, occupation, and area) independently increased the incidence risk of heart failure by 62%, overall [24].

Based on our data, the prevalences of HF were 0.7%, 1.6%, 4.1%, 8.2%, 11.3%, and 21.0% for subjects who were 20–29, 30–39, 40–49, 50–59, 60–69, and ≥70 years of age, respectively. These findings are consistent with previous studies that have demonstrated an increased prevalence of HF following advanced age [12,25]. Researchers from the same region have reported that the prevalences of HF were 0.57%, 3.86%, and 7.55% for individuals who were 25–64, 65–79, and ≥80 years of age, respectively [26]. The current study showed that the prevalence of HF increased with age from 2.0% among 20–49-year-old subjects to 11.0% among 50–87-year-old subjects. Another study showed the prevalence of HF strongly increased with age from 3.0% among 45–54-year-old subjects to 22.0% among 75–83-year-old subjects [26]. 

We observed that HF was more prevalent in men compared to women, despite a significantly higher prevalence of HF in women aged 70 years and older compared to men. These findings agreed with the results of a Chinese study [25]. 

HF is known primarily as a disease of the elderly. However, recent studies have indicated that the HF burden may be increasing in young individuals. Thus, the mean age for HF onset has been declining and the incidence of patients with HF aged below 50 years has increased by two-fold, particularly increasing from 3% to 6% [9]. In a Swedish study that linked national hospital discharge and death registries between 1987 and 2006, HF incidence increased in the last 5-year period by 50% among people aged 18–34 years and 43% among those aged 35–44 years [10]. In our study, the median age for a diagnosis of HF was 50 years, while the mean age at HF diagnosis was 73.7 ± 14.3 years in a UK population aged ≥30 years [27]. Overall, despite the South Asian and African ethnicity groups being significantly younger at HF onset than the Caucasian ethnicity group, they had similar or better cardiovascular risk profiles, which agreed with those previously reported in a younger UK general population [10].

The results of our study showed a highly age-specific prevalence of HF compared to other studies that have mostly included and examined populations aged 45 years and over and used various diagnostic approaches and criteria (see Table 8). Although medical records were reviewed for the definition of HF to identify HF diagnosis according to the Framingham Criteria in the Olmsted County Study [8], subjects in the Rotterdam Study were clinically examined to identify symptoms and signs suggestive of HF (e.g., shortness of breath, ankle oedema, and pulmonary crepitations) [18]. 

There is an ongoing debate regarding the definition of heart failure and there is a lack of a gold standard for assessing the presence of the syndrome in population-based studies. According to the ESC Guidelines on the diagnosis of HF, to establish the presence of heart failure, objective evidence of cardiac dysfunction must be present in addition to symptoms or medication for HF [28]. In the Framingham study, the overall prevalence of HF was 0.7% for those aged between 50 and 89 years, varying between 0.1% and 7.9% with age [28]. In the Rochester study, in 1986, the prevalence of HF in those over 35 years was 1.9%, increasing from 1% to 7.6% with age [29]. 

A recent randomized controlled trial suggested that the most common risk factors for HF were CAD, hypertension, and diabetes mellitus [30]. More specifically, the risk factors highly correlated with HF incidence included poorly controlled diabetes (HbA1c ≥ 8%), uncontrolled hypertension (SBP ≥ 160), and advanced obesity (BMI ≥ 35) [15]. Likewise, our study demonstrated that a previous history of CAD, hypertension, valvular heart disease, obesity, and abnormal alcohol consumption were main risk factors of HF. Because these findings were different compared to the NHANES study [12], we assume that it could be caused by the disparities of living standards and cultural differences.

The present study demonstrated that coronary artery disease (CAD) is the strongest risk factor for the development of HF among other risk factors with a prevalence of 29.1% (*n* = 50) in the total HF population. Secondly, having hypertension was also viewed as a major factor in the progression of HF with a prevalence of 84.9% (*n* = 149) in the HF population in this study. These results were in line with those of former studies such as a cardiovascular health study [31] and a Spanish study (81.8%) [32]. Our analysis also supports that valvular heart disease and obesity are important risk factors in the development of HF. This could be because of a higher incidence of rheumatic heart disease, lack of health education, and the cultural point of view among the Mongolian population.

The strength of our study is the population size which is representative enough for the overall population of Mongolia. Therefore, our study results could be generalizable to patients with and without HF in the general adult population. A limitation is, however, that the symptoms suggestive of HF as well as different disease prevalences (e.g., CHD, COPD, and diabetes) were self-reported, and therefore we were not able to validate this information. Another limitation of this study is that the participants who met the criteria for the clinical diagnosis of chronic HF were not further evaluated through echocardiography and natriuretic peptide testing for a confirmation of the diagnosis. Moreover, the diagnosis of HF was not validated in this study, which increases the risk of observer bias.

## 5. Conclusions

This is the first investigation in Mongolia that describes the prevalence of HF among the general population. The prevalence of HF appears high (4.94%) in the Mongolian population compared with other studies.Our study revealed that coronary heart disease, hypertension, and valvular heart disease are the three foremost risk factors in the development chronic heart failure.

## Figures and Tables

**Figure 1 diagnostics-13-00999-f001:**
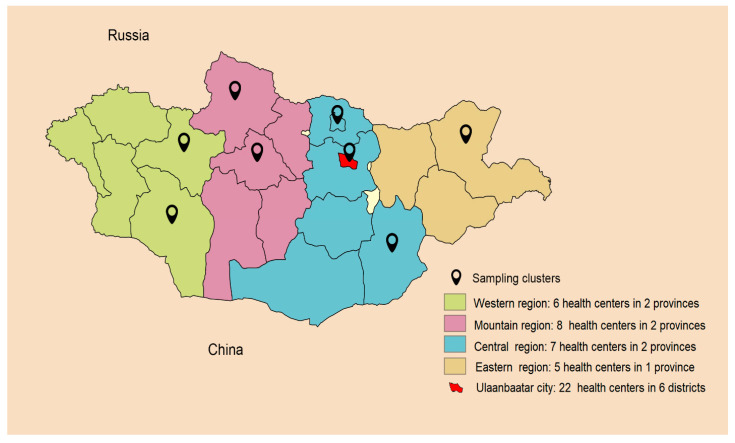
Sampling clusters.

**Figure 2 diagnostics-13-00999-f002:**
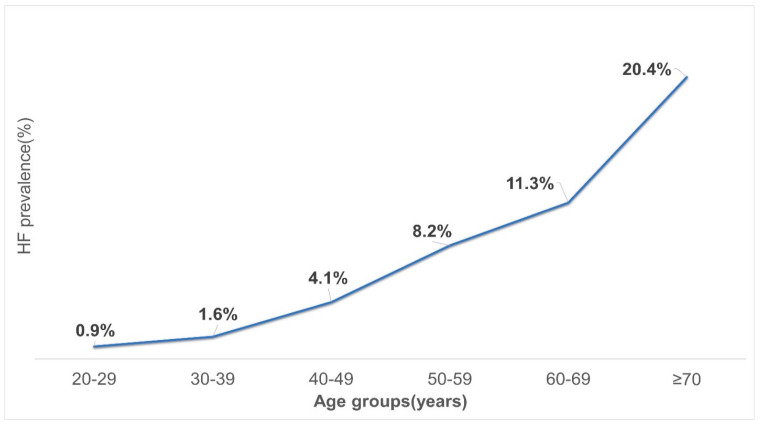
Age–specific prevalence of HF. Note: The vertical axis shows prevalence rate expressed as a percentage. The horizontal axis shows age groups.

**Figure 3 diagnostics-13-00999-f003:**
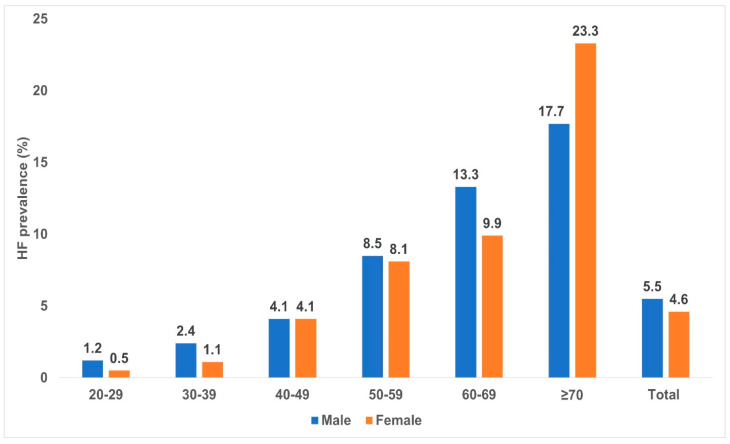
Age–specific prevalence of overall HF by gender.

**Figure 4 diagnostics-13-00999-f004:**
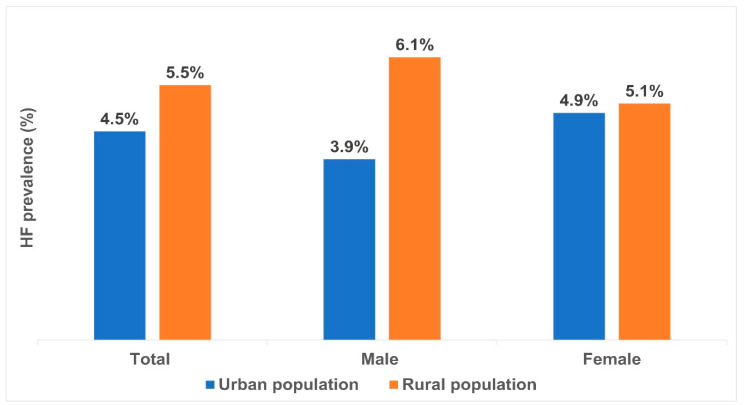
The prevalences of HF in the urban and rural populations. Note: The horizontal axis shows the total, urban, and rural population distribution. The vertical axis shows prevalence rate expressed as a percentage.

**Figure 5 diagnostics-13-00999-f005:**
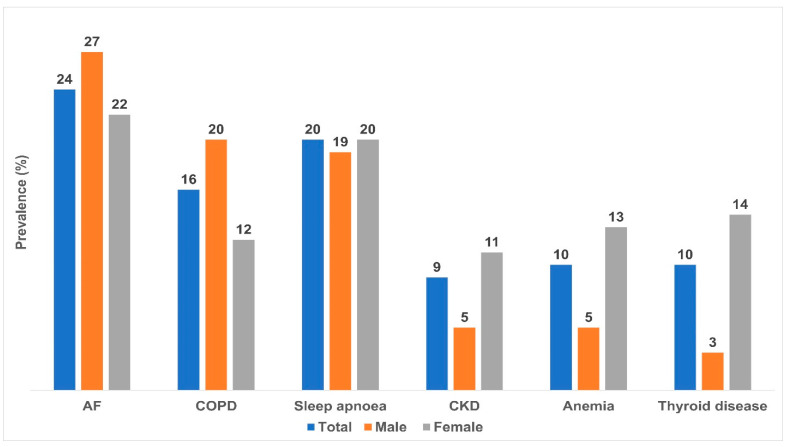
Sex-specific comorbidities in HF. Note: AF, atrial fibrillation; COPD, chronic obstructive pulmonary disease, CKD, chronic kidney disease. The horizontal axis shows comorbidities. The vertical axis shows prevalence rate expressed as a percentage.

**Table 1 diagnostics-13-00999-t001:** Demographic and social characteristics of the study population.

Characteristics	Total Study Population (*n* = 3480)	Male (*n* = 1345)	Female (*n* = 2135)	*p*-Value
Age group				0.484
20–29, *n* (%)	807 (23.2)	286 (21.3)	521 (24.4)	
30–39, *n* (%)	863 (24.8)	335 (24.9)	528 (24.7)	
40–49, *n* (%)	682 (19.6)	268 (19.9)	414 (19.4)	
50–59, *n* (%)	595 (17.1)	236 (17.6)	359 (16.8)	
60–69, *n* (%)	381 (10.9)	158 (11.8)	223 (10.5)	
≥70, *n* (%)	152 (4.4)	62 (4.6)	90 (4.2)	
Education				<0.001
Higher, *n* (%)	1229 (35.3)	418 (31.1)	811 (38.0)	
Medium, *n* (%)	1784 (51.3)	696 (51.7)	1088 (51.0)	
Lower, *n* (%)	467(13.4)	231 (17.2)	236 (11.0)	
Occupation				<0.001
Manual labor, *n* (%)	830 (23.9)	398 (29.6)	432 (20.2)	
Intellectual labor, *n* (%)	1238 (35.7)	457 (34.0)	781 (36.6)	
Unemployed, *n* (%)	1412 (40.4)	490 (36.4)	922 (43.2)	
Marital status				0.432
Married, *n* (%)	2661 (76.5)	1032 (76.7)	1629 (76.3)	
Divorced, *n* (%)	132 (3.8)	45 (3.3)	87 (4.1)	
Unmarried, *n* (%)	687 (19.7)	268 (20.0)	419 (19.6)	0.459
Administrative region				
Urban area, *n* (%)	1686 (48)	641 (47.6)	1045 (49.9)	
Rural area, *n* (%)	1794 (52)	704 (52.3)	1090 (51.1)	

**Table 2 diagnostics-13-00999-t002:** Risk factors for HF according to the age groups.

Variables	Total Subjects (*n* = 3480)	20–29 Years (*n* = 807)	30–39 Years (*n* = 863)	40–49 Years (*n* = 682)	50–59 Years (*n* = 595)	60–69 Years (*n* = 381)	≥70 Years (*n* = 152)	*p*-Value
CAD, *n* (%)	90 (3)	1 (0)	6 (1)	20 (3)	26 (4)	23 (6)	14 (9)	<0.0001
Hypertension, n (%)	1358 (39)	91 (11)	210 (24)	295 (43)	392 (66)	246 (65)	124 (82)	<0.0001
DM, *n* (%)	198 (6)	12 (2)	33 (4)	37 (5)	59 (10)	36 (9)	21 (14)	<0.0001
VHD, *n* (%)	82 (2)	8 (1)	12 (1)	17 (3)	28 (5)	16 (4)	1 (1)	<0.0001
Obesity, *n* (%)	1358 (39)	86 (11)	175 (20)	207 (30)	173 (29)	114 (30)	35 (23)	<0.0001
Smoking, *n* (%)	712 (21)	117 (15)	191 (22)	164 (24)	135 (23)	78 (21)	27 (18)	<0.0001
Abnormal alcohol consumption, *n* (%)	300 (9)	30 (4)	82 (10)	74 (11)	76 (13)	28 (7)	10 (7)	<0.0001

Note: CAD, coronary heart disease; DM, diabetes mellitus; VHD, valvular heart disease.

**Table 3 diagnostics-13-00999-t003:** Risk factors for HF according to the main administrative groups.

Variables	Total Participants *n* = 3480	Population in Urban Area (*n* = 1686)	Population in Rural Area (*n* = 1794)	*p*-Value
Coronary heart disease, *n* (%)	90 (3)	46 (3)	44 (3)	0.609
Hypertension, *n* (%)	1358 (39)	661 (39)	697 (39)	0.831
Diabetes mellitus, *n* (%)	198 (6)	113 (7)	85 (5)	0.012
Valvular heart disease	82 (2)	35 (3)	47 (2)	0.290
Obesity, *n* (%)	790 (23)	350 (21)	440 (25)	0.008
Smoking, *n* (%)	712 (21)	373 (22)	339 (19)	0.018
Abnormal alcohol consumption, *n* (%)	300 (9)	118 (7)	182 (10)	0.001

**Table 4 diagnostics-13-00999-t004:** Comparison of demographic and social characteristics between HF and non-HF groups.

Characteristics	Total Study Population *n* = 3480	Non-HF Group *n* = 3308 (95%)	HF Group *n* = 172 (5%)	*p*-Value
Sex				0.227
Male	1345 (39)	1271 (94.5)	74 (5.5)	
Female	2135 (61)	2037 (95.4)	98 (4.6)	
Age group				<0.0001
20–29	807 (23)	801(24)	6 (4)	
30–39	863 (25)	849 (26)	14 (8)	
40–49	682 (20)	654 (20)	28 (16)	
50–59	595 (17)	546 (16)	49 (28)	
60–69	381 (11)	338 (10)	43 (25)	
≥70	152 (4)	120 (4)	32 (19)	
Average age	41.0 (30.0–54.0)	40.0 (30.0–53.0)	57.0 (49.0–65.8)	<0.0001
Education				<0.0001
Higher	1229 (35)	1198 (36)	31 (18)	
Medium	1784(51)	1695 (51)	89 (52)	
Lower	467(14)	415 (13)	52 (30)	
Occupation				<0.0001
Manual labor work	830 (24)	793 (24)	37 (21)	
Intellectual labor work	1238 (51)	1202 (36)	36 (21)	
Unemployed	1412 (40)	1313 (40)	99 (58)	
Marital status				0.479
Married	2661 (76)	2523 (76)	138 (80)	
Divorced	132 (4)	127 (4)	5 (3)	
Unmarried	687 (20)	658 (20)	29 (17)	

**Table 5 diagnostics-13-00999-t005:** Logistic regression analysis adjusted to age and gender.

Variable	OR	Min Value	Max Value	*p*-Value
Hypertension	4.855	3.127	7.538	<0.0001
CAD	5.117	3.040	8.614	<0.0001
Valvular heart disease	3.872	2.112	7.099	<0.0001
Abnormal alcohol consumption	1.861	1.155	2.998	0.011
Smoking	1.391	0.918	2.109	0.120
Obesity	2.136	1.542	2.959	<0.0001
Diabetes mellitus	1.440	0.865	2.397	0.161

CAD, coronary heart disease

**Table 6 diagnostics-13-00999-t006:** Comparison of risk factors and clinical charactheristics between the HF and non-HF groups.

Variables	Total Study Population (*n* = 3480)	Non-HF Group *n* = 3308 (95%)	HF Group *n* = 172 (5%)	*p*-Value
Risk factors				
Hypertension, *n* (%)	1358 (39)	1213 (37)	145 (84)	<0.0001
Valvular heart disease, *n* (%)	82 (2)	67 (2)	15 (9)	<0.0001
Abnormal alcohol consumption, *n* (%)	300(9)	275 (8)	25 (15)	0.005
Smoking, *n* (%)	712 (20)	667 (20)	45 (26)	0.057
Diabetes mellitus, *n* (%)	198 (6)	178 (5)	20 (12)	0.001
Obesity, *n* (%)	790 (23)	719 (22)	71 (41)	<0.0001
Coronary artery disease, *n* (%)	90 (3)	63 (2)	27 (16)	<0.0001
Clinical charactheristics				
Body mass index, (kg/m^2^)	26.4 ± 5.1	26.2 ± 5.0	28.8 ± 6.2	<0.0001
Heart rate, per minute	80.0 ± 11.1	79.6 ± 10.8	84.9 ± 15.7	<0.0001
Respiratory rate, per minute	18.2 ± 5.1	18.2 ± 5.2	19.2 ± 3.1	0.012
Oxygen saturation, %	96.2 ± 3.7	96.2 ±3.8	95.0 ± 3.1	<0.0001
Systolic blood pressure, mmHg	122.0 ± 19.4	121.2 ± 18.9	137.7 ± 22.4	<0.0001
Diastolic blood pressure, mmHg	78.7 ± 12.6	78.3 ± 12.4	86.2 ± 15.2	<0.0001

**Table 7 diagnostics-13-00999-t007:** Comparison of medications between the HF and non-HF groups.

Medications	Non-HF Group (*n* = 3308)	HF Group (*n* = 172)	*p*-Value
Diuretics, *n* (%)	181 (5.4)	88 (51.2)	<0.0001
RAAS inhibitors, n (%)	210 (6.3)	86 (50.0)	<0.0001
Beta blockers, *n* (%)	190 (5.7)	75 (43.6)	<0.0001
Sacubitril/valsartan, *n* (%)	42 (1.3)	44 (25.6)	<0.0001
Digoxin, *n* (%)	26 (0.8)	26 (15.1)	<0.0001
Ivabradine, *n* (%)	27 (0.8)	22 (12.8)	<0.0001

Note: RAAS, renin-angiotensin-aldosterone system

**Table 8 diagnostics-13-00999-t008:** Prevalence of heart failure in Mongolia compared with other population-based studies.

Age Group	Present Study (Mongolia)	CARLA Study (Germany)	Rotterdam Study (The Netherlands)	Olmsted County Study (USA)
45–54 years	4.3%	3.0%	0.7%	-
55–64 years	9.4%	6.0%	0.7%	1.3%
65–74 years	13.1%	10.4%	2.7%	1.5%
75–84 years	25.7%	22.0%	13%	8.4

## Data Availability

Raw data that support the findings of this study are available from the corresponding author, upon reasonable request.

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
