# Peer review of "The Prevalence and Risk Factors of Chronic Heart Failure in the Mongolian Population"

_diagnostics, 2023, doi:10.3390/diagnostics13050999_

Round 1

Reviewer 1 Report

The manuscript is interesting; however, I am not sure if it falls within the scope of the journal. It seems that a journal focused on public health would be more suitable.

In addition, I would like to have more information on the background pharmacotherapy of enrolled subjects and the correlation with the risk for incident heart failure. 

What is more, it seems that simple characterization as heart failure is not adequate; data regarding sub-type (HFrEF, HFmEF, HFpEF) should be added, if available.

Reviewer 2 Report

The authors investigated the prevalence and risk factors for chronic heart failure in Mongolian population in the manuscript entitled “the prevalence and risk factors of chronic heart failure in Mongolian population”. They reported for the first time the prevalence of heart failure in Mongolian population. They found that hypertension, old myocardial infarction, and valvular heart disease were identified as the three foremost risk factors in the development of heart failure. Several concerns have been raised.

1. Hypertension, old myocardial infarction, and valvular heart disease might be rather the etiology of heart failure: hypertensive heart disease, ischemic cardiomyopathy, and valvular heart failure, instead of risk factors for heart failure.

2. The prevalence of heart failure would be dependent on the definition of heart failure.

3. The etiology of heart failure, such as dilated cardiomyopathy, is unclear.

4. The definition of comorbidity is also unclear.

5. There are many typographs and grammatical errors.  

Round 2

Reviewer 2 Report

There are no further comments.